# How strong is strong? The challenge of interpreting network edge weights

**Zachary P. Neal**  *

Psychology Department, Michigan State University, East Lansing, MI, United States of America

* zpneal@msu.edu

## Abstract

Weighted networks are information-rich and highly-flexible, but they can be difficult to analyze because the interpretation of edges weights is often ambiguous. Specifically, the meaning of a given edge's weight is locally contingent, so that a given weight may be strong for one dyad, but weak for other dyad, even in the same network. I use backbone models to distinguish strong and weak edges in a corpus of 110 weighted networks, and used the results to examine the magnitude of this ambiguity. Although strong edges have larger weights than weak edges on average, a large fraction of edges' weights provide ambiguous information about whether it is strong or weak. Based on these results, I recommend that strong edges should be identified by applying an appropriate backbone model, and that once strong edges have been identified using a backbone model, their original weights should not be directly interpreted or used in subsequent analysis.

**Data Availability Statement:** Data is available from https://osf.io/u8m7n/.

**Funding:** This work was supported by a grant from the National Science Foundation (NSF #2211744). The funders had no role in study design, data

## Introduction

In a weighted network, each edge has an associated weight [1, 2]. These edge weights can be used to encode such information as frequency (e.g., of contact in a social network), intensity (e.g., of traffic in a transportation network), or similarity (e.g., of word pairs in a lexical network). Likewise, edge weights may be measured at ordinal (e.g., often, sometimes, rarely), count (e.g., number of passengers), or interval (e.g., a correlation) levels. Projections of bipartite networks represent a commonly encountered special case of a weighted network [3], where the edge weights record the number of co-occurrences of some artifact for two agents (e.g., the number of bills co-sponsored by two legislators). In short, weighted networks are both highly flexible and information rich.

One key challenge in the analysis of weighted networks is the interpretation of edge weights. Framing the challenge as a question, one must ask '*is an edge with weight w equally strong as another edge in the same network that also has weight w*? Most weighted network analysis methods implicitly assume the answer is yes, and indeed there are cases where edge weights have a universally consistent interpretation. Consider, for example, the case of a contact frequency network used to study disease transmission: Exposure to one possible source of infection for 15 minutes is the same as exposure to another source for 15 minutes. However, in many other cases the interpretation of an edge's weight is more locally contingent. Consider,

collection and analysis, decision to publish, or preparation of the manuscript.

**Competing interests:** The authors have declared that no competing interests exist.

for example, the case of an airline traffic network: Observing 1,000,000 passengers fly between New York and Los Angeles is unremarkable, but observing 1,000,000 passengers fly between Scranton and Stockton is noteworthy. In this example, an edge weight of 1,000,000 has an ambiguous interpretation because for one dyad it is weak, while for another dyad in the same network it is strong.

A large family of backbone and sparsification methods offer ways to distinguish weak and strong edges based on their weights, but conditioned on local features of the network [4]. In this paper, I use these methods to investigate the ambiguity of edge weights. I begin by introducing backbone methods, then illustrating the issue of edge weight ambiguity using a weighted network of US air traffic and a bipartite projection of US Senators' bill sponsorships. I then formally investigate the issue in a corpus of 54 empirical weighted networks and projections of 54 synthetic bipartite networks. I find that although edges deemed strong by a backbone model have larger weights than edges deemed weak *on average*, a large fraction of edges have an ambiguous weight that is sometimes judged as strong and other times as weak. Based on these findings, I conclude by recommending that strong edges should be identified by applying an appropriate backbone model, and not by direct inspection of edge weights. I also recommend that the original weights of edges judged to be strong by a backbone model should not be directly interpreted or used in subsequent analyses.

## Identifying 'strong' edges

Given the challenges of analyzing and visualizing weighted networks [4], many methods have been proposed for identifying strong edges in weighted networks [5–22] or projections of bipartite networks [23–28]. The sparser subgraph composed only of these strong edges is known as the 'backbone' of the network, and therefore these methods are known as backbone extraction or sparsification methods.

These methods differ in their computational details, but comparative evaluations suggest that they yield broadly similar backbones [12, 21, 28]. I focus primarily on two methods that are computationally fast and have been widely-adopted: the Disparity Filter (DF) [5] for extracting backbones from weighted networks, and the Stochastic Degree Sequence Model (SDSM) [23] for extracting backbones from bipartite projections. In this section, I introduce these methods and illustrate their application. In the results section, I present a sensitivity test that examines the robustness of the main results to some other widely-used backbone methods.

## Weighted networks

Many weighted networks exhibit a multiscalar structure in which the weights are 'broadly distributed, spanning several orders of magnitude" [5]. This can mean that some parts of the network contain systematically stronger edges than other parts, and that there is not a single characteristic scale for the edge weights. Fig 1A illustrates the continental US air traffic network in 2019, where edges are weighted (and drawn thicker/darker) by the number of passengers traveling on each route, and offers an example of such a multiscalar weighted network. In this case, routes between large cities and hub airports carry many more passengers (e.g., 3.3M between JKF in New York and LAX in Los Angeles) than routes between hub airports and regional 'spoke' airports (e.g., 10 between MIA in Miami and ROA in Roanoke, Virginia). Although hub-spoke routes have smaller weights, they are often just as important as the higher weight hub-hub routes for transporting passengers and maintaining the connectivity of the network. That is, the absolute weight of an edge (here, the absolute number of passengers) provides ambiguous information about the edge's strength in the network.

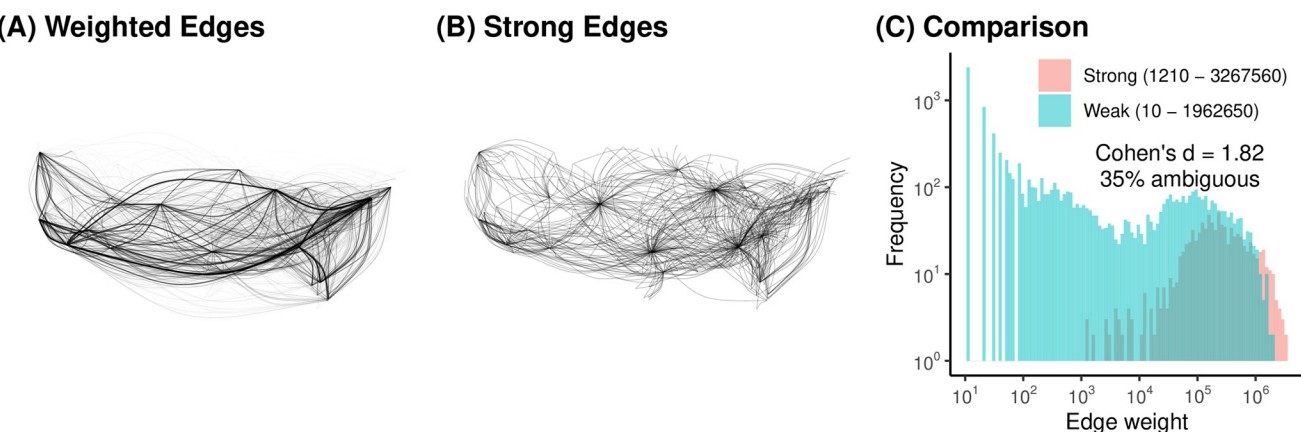

**Fig 1.** The 2019 US air traffic network with (A) all edges weighted by passenger volume, and (B) only edges deemed strong by DF with $\alpha$ = 0.001. (C) The weights of strong and weak edges overlap substantially.

To overcome this issue, the fast and widely-used Disparity Filter (DF) backbone model distinguishes weak and strong edges by taking into account the network's multiscalar characteristics when evaluating the edge weights [5]. It treats an edge's weight as strong if it contributes more to a node's total degree than if edge weights were randomly distributed. By evaluating an edge's weight with respect to a given node's degree, DF implicitly asks "Is this edge strong *for this node*", thereby making locally contingent judgements of the strength of edges. The formal mathematical specification of DF is provided by [5], and it is implemented in the R `backbone` package [4]. Fig 1B illustrates the DF backbone extracted from the original weighted network in Fig 1A, using an $\alpha$ = 0.001 threshold of statistical significance. The DF backbone preserves some (but not all) high-volume routes, and also preserves some (but not all) low-volume routes, thereby capturing the network's hub-and-spoke structure.

Fig 1C plots (on a double-log scale) the frequency of edges weights among edges deemed strong (blue) or weak (pink) by DF. The strong edges ranged in weight from 1210 to 3,267,560 ($M$ = 400, 128.6, $SD$ = 482765.9), while the weak edges ranged in weight from 10 to 1,962,650 ($M$ = 43327.2, $SD$ = 142967.9). Cohen's $d$ is a widely-used standardized measure of the magnitude of difference in the mean of two samples [29]. Here, $d$ = 1.82, which corresponds to a very large difference. That is, the edges that DF views as strong and retains in the backbone have larger weights, on average, than the edges DF views as weak and discards.

Such a large $d$ might lead some to conclude that larger weights indicate stronger edges, while smaller weights indicate weaker edges. However, closer inspection of the edge weights plotted in Fig 1C shows that such a conclusion is incorrect. There are many cases where an edge with an objectively large weight was deemed weak. For example, observing 1,962,650 passengers flying between ATL in Atlanta and LAX in Los Angeles is unremarkable because ATL is the largest airport in the world. There are also many cases where an edge with an objectively small weight was deemed strong. For example, observing just 1210 passengers flying between PDX in Portland, Oregon and PDT in Pendleton, Oregon is noteworthy because these are both small airports; this route exists due to a federal subsidy under the Essential Air Service program.

These two examples point to the ambiguity of information conveyed by edge weights in a weighted network. Strong edges do not necessarily have large weights, and weak edges do not necessarily have small weights. The magnitude of this ambiguity can be captured by the fraction of edges whose weights lie within the range of ambiguous weights (i.e. the smallest weight

deemed strong, and the largest weight deemed weak). In this network, 35% of all edges lie within this range. In practical terms, this means that for one-third of the edges, the edge weight does not provide clear information about whether the edge is strong.

## Bipartite projections

Given the indicence matrix of a bipartite network **B** composed of two types of nodes ('agents' and 'artifacts'), such that edges can exist only between an agent and an artifact, the adjacency matrix of its projection **P** can be obtained as $\mathbf{P} = \mathbf{BB}^{T}$. The projection is a special case of a weighted network, where the edge weight between two agents captures their number of shared artifacts. Fig 2A illustrates the bill co-sponsorship network among US Senators during the 115[th] (2017-2018) session. The edges are weighted (and drawn thicker/darker) by the number of bills (the artifacts) that two senators (the agents) both sponsored, and the nodes are colored by the Senators' political party affiliation (red = Republican, blue = Democrat, green = Independent). One common challenge in studying bipartite projections is that the projection function induces many edges, leading to a very dense network [3]. In this case, every Senator is connected to every other Senator because all pairs sponsored at least one bill together, leading the network to appear more cohesive (i.e., bipartisan) than it actually is.

Backbone models designed specifically for bipartite projections are similar to other backbone models, but are able to exploit the additional information available in the original bipartite network. Specifically, such models can consider the commonness or rarity of individual artifacts, recognizing that sharing a common artifact (e.g., both sponsoring a universally popular bill) is unremarkable, while sharing a rare artifact (e.g., both sponsoring a widely disliked bill) is noteworthy. The fast and widely-used Stochastic Degree Sequence Model (SDSM) [23] is one such backbone model. It treats an edge's weight as strong if it is larger than expected by comparison to projections of random bipartite networks in which each agent and each artifact had the same expected degree. This approach means that SDSM implicitly asked "Is this edge strong *for these nodes, and given their particular shared artifacts*", thereby making more informed judgements of the strength of edges. The formal mathematical specification of SDSM is provided by [23], and it is implemented in the R `backbone` package [4]. Fig 2B illustrates the SDSM backbone extracted from the original weighted network in Fig 2A, using an $\alpha = 0.01$ threshold of statistical significance. The SDSM backbone preserves some (but not all) partnerships characterized by a large number of co-sponsorships, but also preserves some (but

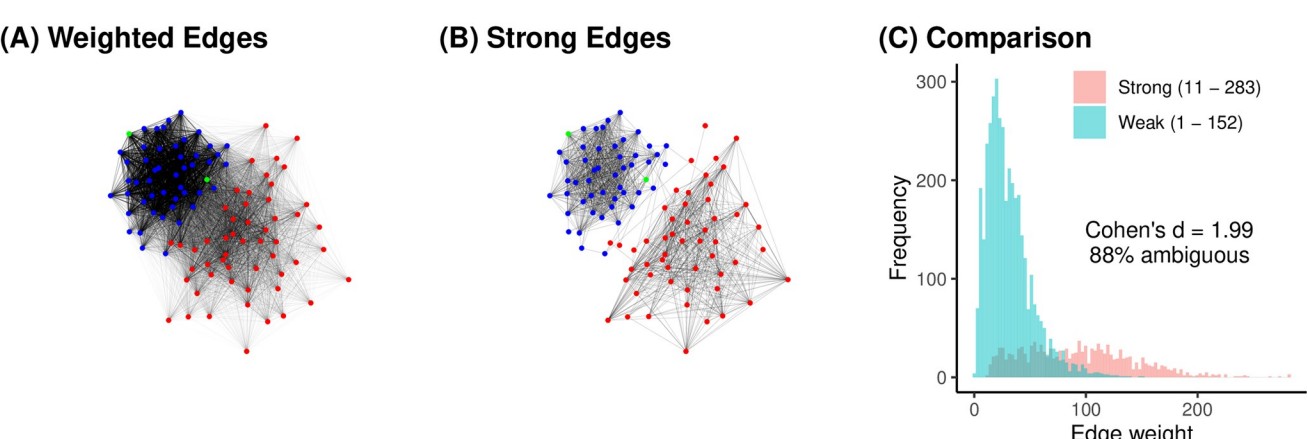

**Fig 2.** The 115[th] US Senate bill co-sponsorship network with (A) all edges weighted by number of co-sponsored bills, and (B) only edges deemed strong by SDSM with $\alpha = 0.01$. (C) The weights of strong and weak edges overlap substantially.

not all) partnerships characterized by a small number of co-sponsorships, thereby more clearly capturing the network's partisan and polarized structure.

Fig 2C plots the frequency of edges weights among edges deemed strong (blue) or weak (pink) by SDSM. The strong edges ranged in weight from 11 to 283 ($M$ = 92.6, $SD$ = 49.7), while the weak edges ranged in weight from 10 to 1,962,650 ($M$ = 31.3, $SD$ = 20.8). Here, very large $d$ = 1.99 indicates that the edges that SDSM views as strong and retains in the backbone have larger weights, on average, than the edges SDSM views as weak and discards.

As before, the large $d$ could lead to the erroneous conclusion that larger weights indicate stronger edges, while smaller weights indicate weaker edges. However, there are many cases where an edge with an objectively large weight was deemed non-significant. For example, observing 152 co-sponsorships between Sen. Merkley [D-OR] and Sen. Klobuchar [D-MN] is unremarkable; they sponsored a whopping 401 and 492 bills, respectively, making at least some co-sponsorships inevitable. There are also many cases where an edge with an objectively small weight was deemed significant. For example, observing just 11 co-sponsorships between Sen. McConnell [R-KY] and Sen. Corker [R-TN] is noteworthy; as the majority leader, Sen. McConnell sponsored relatively few bills (51), making any co-sponsorships with him unusual.

These two examples point to the ambiguity of information conveyed by edge weights in a bipartite projection. As in traditional weighted networks, strong edges do not necessarily have large weights, and weak edges do not necessarily have small weights. Again, the magnitude of this ambiguity can be captured by the fraction of edges whose weights lie within the range of ambiguous weights. In this network, 88% of all edges lie within this range. In practical terms, this means that for nearly all the edges, the edge weight does not provide clear information about whether the edge is strong.

## Methods

### Data

I investigate the ambiguity of edge weights in a total of 110 networks obtained from three sources.

First, the US air traffic and US Senate networks described in the prior section come from the documentation for the R `backbone` package's documentation [4]. The air traffic network was originally generated using data from the US Bureau of Transportation Statistics' Airline Origin and Destination Survey, which contains a 10% random sample of all domestic airline tickets in 2019, and closely resembles the network originally used to develop DF [5]. The Senate network was originally generated using data from the US Government Publishing Office [30], and closely resembles the network originally used to develop SDSM [23].

Second, 54 empirical weighted networks come from a public corpus of such networks originally assembled to investigate the 'Strength of Weak Ties' theory [31]. These networks were obtained from seven large network data repositories: SocioPatterns [32], University of California Irvine Network Data Repository [33], Netzschleuder [34], the KONECT Project [35], the Index of Complex Networks [36], and two `networkdata` libraries for R [37, 38]. They include networks from diverse contexts (e.g., school, work), with diverse types of edges (e.g., communication, friendship) whose weighted were measured in diverse ways (e.g., frequency, intensity). Fig 3 summarizes the setting, size, density, and clustering coefficient of the networks contained in this corpus.

Third, because there is not an existing corpus of bipartite projections and because network repositories contain relatively few bipartite networks, 54 synthetic bipartite networks were generated, then projected. These networks were constructed to systematically vary four characteristics known to be important for bipartite projections' structure [28, 39]: size (100

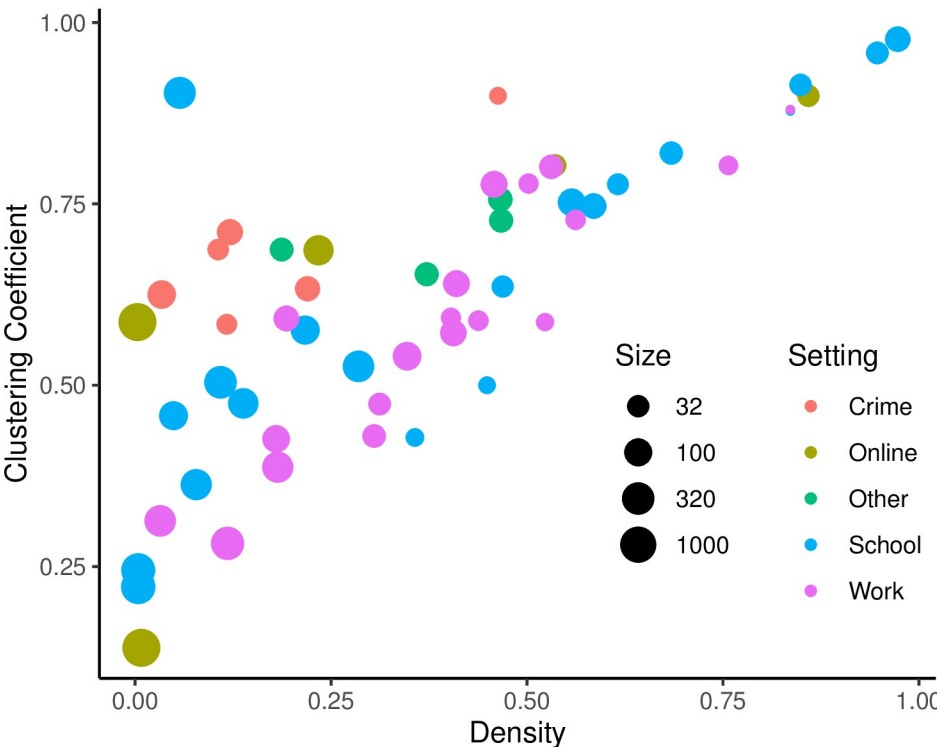

**Fig 3. Setting, size, density, and clustering coefficient of 54 empirical weighted networks extracted from the corpus provided by [31].**

agents × 500 artifacts, 200 × 2000, and 300 × 6000), agent degree distribution (left-tailed, right-tailed, normal), artifact degree distribution (left-tailed, right-tailed, normal), and community partitions (weak, strong).

All data described in this paper and needed to replicate the results is available at https://osf.io/u8m7n/.

## Analysis

In the primary analysis, I extract the backbone from each network using DF (for weighted networks) or SDSM (for bipartite projections), with $\alpha = 0.05$. As illustrated above in the air traffic and senate examples, I then compute both Cohen's $d$ and the fraction of edges with ambiguous weights. Finally, I plot these two values to understand their relationship, and to examine the magnitude of ambiguity in edge weights.

The primary analysis uses the most widely-adopted backbone models and the commonly-used $\alpha = 0.05$ threshold for statistical significance. To evaluate the robustness of the primary results, I conduct three sensitivity tests. First, I repeat the primary analysis using the more stringent $\alpha = 0.01$, which yields sparser backbones. Second, I repeat the primary analysis using the more liberal $\alpha = 0.2$, which yields denser backbones. Third, I repeat the primary analysis, but use alternative backbone models: Locally Adaptive Network Sparsification (LANS, for weighted networks) [6], and the Fixed Row model (FR, for bipartite projections) [24]. These backbone models are less commonly used than DF or SDSM, but have also been validated and often yield similar results.

The code necessary to replicate the examples above, and the results reported below, is available at https://osf.io/u8m7n/.

## Results

### Primary analysis

Fig 4 plots Cohen's *d* (x-axis) and the fraction of edges with ambiguous weights (y-axis) in each of 110 networks. Weighted networks are shown as red circles, while bipartite projections are shown as blue squares; the two networks used as examples above are plotted using larger symbols. As expected, there is an S-shaped relationship between these two values. When *d* is larger, reflecting a larger difference in the average edge weight of strong and weak edges, the meaning of edge weights is less ambiguous. In contrast, when *d* is smaller, reflecting a smaller difference in the average edge weight of strong and weak edges, the meaning of edge weights is more ambiguous. Three patterns in these results are noteworthy.

First, *d* is always very large given the conventional thresholds used to interpret this statistic [29]. This indicates that, on average, edges deemed to be strong by a backbone model have much larger weights than edges deemed to be weak.

Second, the percent of edges with ambiguous weights is also often large. Although on average significant edges have larger weights, many edges' weights still provide ambiguous information about the edge's strength. In most of these networks, at least 20% of edges' weights have an ambiguous interpretation, while in many cases more than half (and sometimes all) the edges' weights have an ambiguous interpretation.

Third, on average, more edge weights in bipartite projections are ambiguous than in weighted networks. Specifically, 20-40% of edge weights in weighted networks have an ambiguous interpretation, while 60-100% of edge weights in bipartite projections have an ambiguous interpretation.

### Sensitivity analysis

The results of the primary analysis could be an artifact of the $\alpha$ thresholds or models used to extract the backbones. To investigate this possibility, Fig 5 repeats the analysis using (A) a

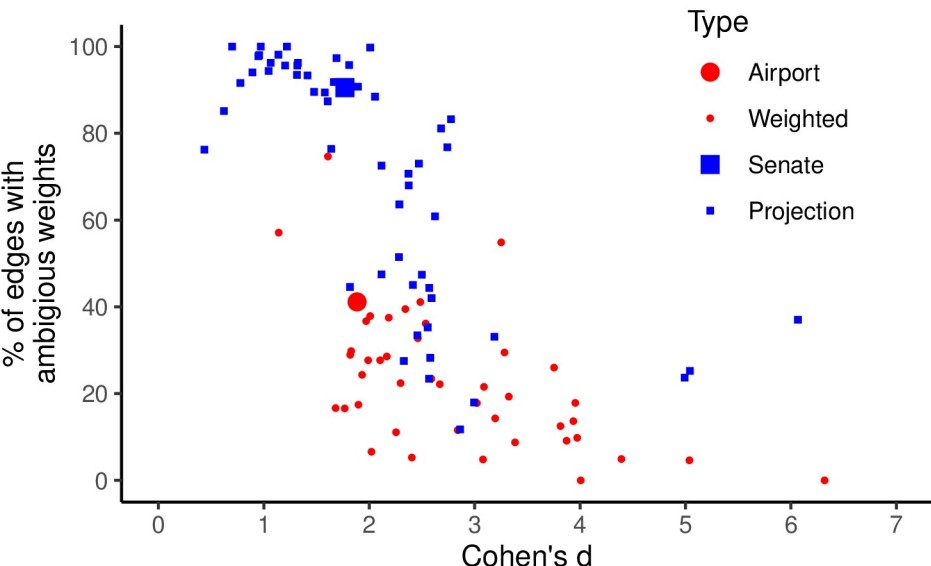

**Fig 4. The standardized effect of statistical significance on edge weight (x-axis) and percent of edges whose weights are ambiguous (y-axis) for a corpus of empirical weighted networks (red) and projections of simulated bipartite networks (blue), when backbones are extracted using DF (weighted) or SDSM (projection) with $\alpha = 0.05$.**

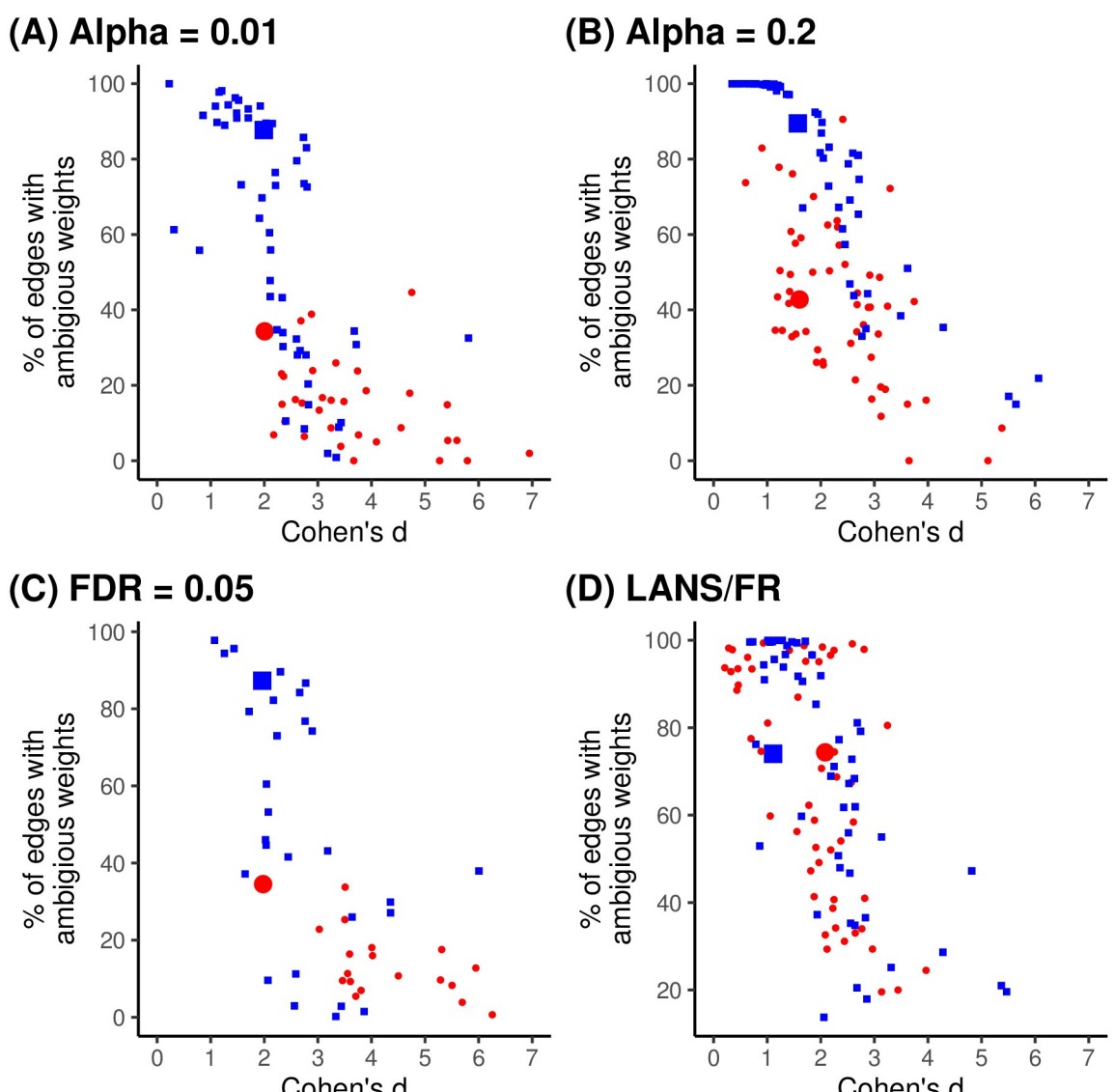

**Fig 5. The standardized effect of statistical significance on edge weight (x-axis) and percent of edges whose weights are ambiguous (y-axis) for a corpus of empirical weighted networks (red) and projections of simulated bipartite networks (blue) when backbones are extracted using DF and SDSM with (A) $\alpha = 0.01$, (B) $\alpha = 0.2$, (C) FDR = 0.05, or (D) using LANS and FR with $\alpha = 0.05$.**

more stringent $\alpha$ to extract sparser backbones, (B) a more liberal $\alpha$ to extract denser back-bones, (C) a false discovery rate (FDR) of 0.05 that controls for the fact that extracting a back-bone involved performing a large number of independent hypothesis tests, and (D) alternate models to extract different backbones. All of the patterns observed in the primary analysis are replicated in these sensitivity analyses, suggesting that the results are robust to changes in the significance threshold and backbone model.

## Discussion

Weighted networks are information rich because they can encode information about the frequency, intensity, or similarity of a relationship. They are also highly flexible because they can

encode this information at different levels (e.g., ordinal, count, interval) in different contexts (e.g., infrastructure networks, social networks, etc.). Despite their benefits, weighted networks can be difficult to understand because the interpretation of edges' weights is often ambiguous. Specifically, it can be ambiguous whether an edge with a weight $w$ should be regarded as weak or strong. The challenge lies in the fact that, even within a single network, one edge with weight $w$ may be strong, while another edge with weight $w$ may be weak, because the meaning of edge weights is locally contingent. Backbone models offer a way to overcome this challenge by distinguishing strong from weak edges based not only on their weights, but also on other local information.

I use backbone models to examine the ambiguity of edge weights. Specifically, in a corpus of 110 networks, I compare the weight of edges deemed strong by a backbone model to the weight of edges deemed weak by a backbone model. On average, strong edges have larger weights, while weak edges have smaller weights. However, despite this intuitive general pattern, individual edges' weights frequently provide ambiguous information about whether the edge is strong or weak. In weighted networks 20-40% of edges weights provide ambiguous information about whether the edge is strong or weak, while in weighted bipartite projections 60-100% of edges weights provide ambiguous information about strength.

This study has a number of strengths, including evaluating edge weight ambiguity in a large and diverse corpus of networks, using publicly-available data and replicable code, and testing the robustness of conclusions through sensitivity tests. At the same time, the findings must be interpreted in light of some limitations. First, these analyses focus only the most widely used and well-known backbone models, but do not examine the dozens of backbone models that exist in the literature [5–28]. However, the provided code allows researchers to replicate these analyses for any backbone model of interest. Second, these analyses demonstrate a consistent pattern in a diverse corpus of networks, but cannot necessarily draw conclusions about all weighted networks. However, the provided code allows researchers to compute both $d$ and the percent of ambiguous weights for any network of interest. Finally, although these analyses demonstrate that many edge weights in weighted networks provide ambiguous information about the edges' strength, there is significant variation in the magnitude of this ambiguity across networks. In general, more edge weights are ambiguous in bipartite projections than in conventional weighted networks, but within these types of networks, investigating the properties that contribute to the magnitude of ambiguity is an important direction for future research.

Despite these limitations, which point to directions for future research, the results support two recommendations for analyzing weighted networks. First, *strong edges should be identified by applying an appropriate backbone model, and not by direct inspection of edge weights.* As these results illustrate, deciding which edges are weak or strong by directly inspecting their weights would often lead to incorrect classifications. Second, *once strong edges have been identified using a backbone model, their original weights should not be directly interpreted or used in subsequent analysis.* As these results illustrate, an edge's weight often provides little information about its strength in the network. Therefore, even among edges deemed strong by a backbone model, edges' weights will vary is ways that are not directly related to their strength.

## Author Contributions

**Conceptualization:** Zachary P. Neal.

**Data curation:** Zachary P. Neal.

**Formal analysis:** Zachary P. Neal.

**Funding acquisition:** Zachary P. Neal.

**Investigation:** Zachary P. Neal.

**Methodology:** Zachary P. Neal.

**Software:** Zachary P. Neal.

**Visualization:** Zachary P. Neal.

**Writing – original draft:** Zachary P. Neal.

**Writing – review & editing:** Zachary P. Neal.

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
