## [Decision Letter · Decision Letter 0]

3 Sep 2024

PONE-D-24-28454How strong is strong? The challenge of interpreting network edge weightsPLOS ONE

Dear Dr. Neal,

Thank you for submitting your manuscript to PLOS ONE. After careful consideration, we feel that it has merit but does not fully meet PLOS ONE’s publication criteria as it currently stands. Therefore, we invite you to submit a revised version of the manuscript that addresses the points raised during the review process.

We look forward to receiving your revised manuscript.

Kind regards,

Hocine Cherifi

Academic Editor

PLOS ONE

Journal Requirements:

3. Thank you for stating the following financial disclosure: National Science Foundation, #2211744

Reviewers' comments:

Reviewer's Responses to Questions

**Comments to the Author**

1. Is the manuscript technically sound, and do the data support the conclusions?

Reviewer #1: Yes

2. Has the statistical analysis been performed appropriately and rigorously? 

Reviewer #1: No

3. Have the authors made all data underlying the findings in their manuscript fully available?

Reviewer #1: Yes

4. Is the manuscript presented in an intelligible fashion and written in standard English?

Reviewer #1: Yes

5. Review Comments to the Author

Reviewer #1: The paper is well-organized and well-written. However, I have the following suggestions:

1. I recommend adding a figure that briefly describes the data to demonstrate that it covers a wide range of networks. For instance, you can plot the networks with density on the x-axis, clustering coefficient on the y-axis, each network category (transportation, social, biological, etc.) represented by different shapes (square, circle, triangle, etc.), and the point size representing the number of nodes.

2. The analysis is conducted using different statistical significance levels. However, the methods employed involve multiple significance testing, which necessitates corrections such as Bonferroni or False Discovery Rate (FDR). The author should consider performing the analysis with and without multiple testing corrections to observe their effect on Cohen's d and the percentage of ambiguity.

Additionally, there are minor typos:

1. Line 100: "These two examples point to the ambiguity of information conveyed by edge weights in a weighted network." should be "This example points...".

2. Line 134: "The formal mathematical specification of DF is provided by" should be "The formal mathematical specification of SDSM is provided by".

Once these points are addressed, I believe the manuscript will make a significant contribution to the field of Network Filtering, as the proposed measure (ambiguity) can help assess different methods in the literature and choose the best one.

6. PLOS authors have the option to publish the peer review history of their article (what does this mean?). If published, this will include your full peer review and any attached files.

Reviewer #1: No

---

## [Editor Report · Decision Letter 1]

17 Sep 2024

How strong is strong? The challenge of interpreting network edge weights

PONE-D-24-28454R1

Dear Dr. Neal,

We’re pleased to inform you that your manuscript has been judged scientifically suitable for publication and will be formally accepted for publication once it meets all outstanding technical requirements.

Kind regards,

Hocine Cherifi

Academic Editor

PLOS ONE
---

## [Editor Report · Acceptance letter]

23 Sep 2024

PONE-D-24-28454R1 

PLOS ONE

Dear Dr. Neal, 

I'm pleased to inform you that your manuscript has been deemed suitable for publication in PLOS ONE. Congratulations! Your manuscript is now being handed over to our production team.

Kind regards, 

on behalf of

Professor Hocine Cherifi 

Academic Editor

PLOS ONE